# Quantitative Ultrasound Examination (QUS) of the Calcaneus in Long-Term Martial Arts Training on the Example of Long-Time Practitioners of Okinawa Kobudo/Karate Shorin-Ryu

**DOI:** 10.3390/ijerph20032708

**Published:** 2023-02-03

**Authors:** Wojciech M. Glinkowski, Agnieszka Żukowska, Bożena Glinkowska

**Affiliations:** 1Center of Excellence “TeleOrto”, Telediagnostics and Treatment of Disorders and Injuries of the Locomotor System, Department of Medical Informatics and Telemedicine, Medical University of Warsaw, 00-581 Warsaw, Poland; 2Polish Telemedicine and eHealth Society, 03-728 Warsaw, Poland; 3Gabinet Lekarski, 03-728 Warsaw, Poland

**Keywords:** quantitative ultrasound, QUS, bone, sports performance, sports, martial arts, osteoporosis prevention

## Abstract

Quantitative research of bone tissue related to physical activity (PA) and sport has a preventive dimension. Increasing the parameters of bone tissue strength, especially reaching the maximum value of peak bone strength in childhood, adolescence, and early adulthood due to practicing sports can contribute to maintaining bone health throughout life. Practicing martial arts (tai chi, traditional karate, judo, and boxing) can effectively improve the quality of bone and reduce the risk of falls and fractures. The study aimed to evaluate the calcaneus bones among Okinawa Kobudo/Karate Shorin-Ryu practitioners using the quantitative ultrasound method as an indicator for assessing bone fracture risk. Forty-four adult martial arts practitioners with a mean age of 36.4 participated in this study. Quantitative ultrasound (QUS) with a portable Bone Ultrasonometer was used in this study. Speed of sound (SOS), broadband ultrasound attenuation (BUA), and the stiffness index (SI) were measured. Subjects were assigned to two groups of black and color belts, according to the advancement in Kobudo/karate practice. The measurements of the SI, BUA, SOS, T-score, and Z-score were significantly higher in subjects from the advanced, long-term practice (black belts) (*p* < 0.05). The long-term martial arts training in traditional karate and Kobudo significantly impacts the parameters of the calcaneus quantitative ultrasound measurements. Significantly higher bone density was observed among the black belt holders. Long-term practice subjects achieved results far beyond the norm for their age groups. Further studies using non-invasive methods of bone quantification are needed to determine the specific conditions for preventing osteoporosis through physical activity, sports, and martial arts, particularly the duration of the activity, the magnitude of loads, and other related factors.

## 1. Introduction

The bone tissue strength and density measurements are primarily associated with osteoporosis diagnostics [1,2,3,4,5,6,7]. Osteoporosis has been defined as a skeletal disorder characterized by impaired bone strength predisposing those with it to an increased risk of fractures [6]. A more recent definition of osteoporosis combines impaired bone strength and fractures, low bone density, the deterioration of bone tissue, and disrupted bone microarchitecture [7]. Bone mineral density (BMD) and bone quality refer to the strength of the bone. BMD examined with the dual-energy X-ray absorptiometry (DXA) method is considered as standard [6,7,8]. Quantitative ultrasonometry (QUS) has been recognized as an osteoporosis screening tool [4,9,10,11,12]. However, among active adults and generally healthy people, the disease state of osteoporosis can rarely be suspected. Therefore, the importance of bone tissue testing in sports has mainly a preventive connotation. The studies of bone tissue quality among physically active people usually emphasize the prophylactic aspect of increasing the parameters of bone tissue strength [13,14]. For subjects actively practicing sports, higher initial parameters are recorded, indicating higher bone density and strength, compared to group peers not practicing sports [15,16,17,18,19,20,21,22]. Physical exercise has been shown to positively affect BMD, especially regarding the impact of weight-bearing activity [23,24,25,26,27,28,29,30]. Physical activity (PA) is commonly recommended in subjects at risk for osteoporosis due to its potential to improve balance. Karate, among other martial arts, is considered a form of PA producing better postural effects, including stability [31], strength [32], and kinematics [33]. Little is known about how Kobudo, simultaneously with karate practice, influences bone strength or density of the lower extremities. A few articles in the literature show that practicing martial arts can effectively improve bone quality and reduce fracture risk and falls [34,35,36,37,38,39,40,41]. The effectiveness of bone quality improvement was described for practicing tai chi [42,43,44,45,46,47,48,49], karate [18,50,51,52,53], judo [54,55,56], and boxing [17,57,58,59]. The study aimed to analyze the long-term impact of martial arts practice based on Okinawa Kobudo/Karate Shorin-Ryu practitioners on the skeletal status of lower extremities as a bone quality and fracture risk indicator using the quantitative ultrasound method.

## 2. Materials and Methods

A group of 44 healthy volunteers, participants of the internship, was included in the study. Subjects were assigned to two groups according to the advancement in Kobudo/Karate practice: group 1—highly advanced (black belts), and the second group—(moderately advanced) color belts. The study protocol adhered to the ethical standards of the Helsinki Declaration, and the Medical University of Warsaw Bioethics Committee approved the study (No. KB 22/2012, issued 17 January 2012). The study was conducted under the umbrella of the broader project concerned with civilization aspects of the locomotor system shortly after obtaining the protocol approval. The events of advanced training are relatively unique and organized only a few times a year with the participation of advanced black belts (long-term practitioners). Some color belts were allowed to take part in this event. The main criterion for stage participation is the advancement in martial art. In addition, only the presented number of martial arts practitioners agreed to participate in non-invasive studies. The research was carried out in particular circumstances where participants from many countries came for the training sessions under the supervision of a top master, and then the number of people resulted from time constraints and limited availability of such people in a given area. Measurements were performed during the single International Open Stage of Okinawa Kobudo, conducted in Warsaw, 17–18 November 2012. Polish Shorin-Ryu Karate Kobudo Union (a member of the World Oshukai Okinawa Shorin-Ryu Karate Do Kobudo Federation) organized the event [60]. The federation represents the traditional martial art of Okinawan Karate and Kobudo. Kobudo is a martial art using Okinawan weapons, determined based on kata and their interpretation taught “hand in hand”. The World Oshukai Dento Okinawa Shorin-Ryu Karate Do Kobudo Federation represents traditional martial arts from OKINAWA: Karate and Kobudo [61]. The World Oshukai Dento Okinawa Kobudo is based on the style of the Matayoshi family and has its dojos on five continents. The principal weapons used in Okinawa Kobudo include Bo, Sai, Tonkuwa, Nunchaku, Jo, Sansetsukon, Nunti, Eku, Kama, Kue, Timbe, Suruchin, Tetcho, and Tekko [61]. Most Kobudo practitioners regularly practice karate. Most martial arts exercises practiced are barefoot. The bone status was assessed using quantitative ultrasound. Quantitative ultrasonometry (QUS) has been a bone health measurement method for 40 years [10,62,63,64].

It can be assumed as the closest to determining bone strength because it allows calculation of the local strength of the assessed material non-invasively. An additional advantage of ultrasound is its safety because it uses no harmful ionizing radiation. Two probes are used in QUS devices: a transmitting and a receiving one [65]. The calcaneus is the most common place for QUS measurements. It is used because it has a relatively thin layer of compact bone surrounding the trabecular bone’s interior [9,62,63,64,66]. Calcaneal QUS measurement is recognized as an indicator of bone health, according to the International Society of Clinical Densitometry (ISCD), because more research has been conducted on the calcaneus compared to other bone segments [67]. The speed of sound (SOS) measured in meters per second (m/s) and broadband ultrasonic attenuation (BUA) measured in dB/MHz are parameters commonly generated by QUS. Broadband ultrasound attenuation (BUA, in decibels per megahertz) and speed of sound (SOS, in meters per second), and SI were measured for the right heel bone using a portable Achilles Express II Bone Ultrasonometer (GE Lunar Corporation, Madison, WI, USA).

Statistical analysis was performed using statistical software TIBCO Statistica 13.3; TIBCO Software, Inc.: Palo Alto, CA, USA, 2017). A *p*-value of <0.05 was considered statistically significant.

## 3. Results

Forty-four martial artists signed informed consent for this study. The mean age of all the subjects was 36.4 years (from 20 to 68; std 10.7). Twenty-four participants were assigned to the black belt group (master’s degrees DAN—a median of 4 (from 1 to 9), and 20 participants to the color belt group—students’ degree holders kyu—a median of 4 (1–6). The black belt subjects had a mean age of 39.9 years (std. 11.3), and color belts 32.1 years (std. 8.4). The participants represented Poland, France, the Czech Republic, Slovakia, Norway, and Hungary. In the study, 42 participants were Caucasians, and two were Asians. The interview confirmed no glucocorticoid intake, history of fragility fractures, or smoking. The female participants (seven) declared no hormonal replacement therapy.

The mean weekly activity declared by all participants was 10.9 h (from 3 to 70 std 9.9). As expected, subjects from the black belt group confirmed a significantly longer overall duration of their practice (*p* < 0.05). The SI, BUA, SOS, T-score, and Z-score were significantly higher in the black belt group (*p* < 0.05).

As expected, the black belt practitioners were significantly older than the color belt holders (Student’s *t*-test: F (1, 42) = 6.4419, *p* = 0.015). The mean values of the quantitative ultrasound measurements of bones were significantly higher in the black belt group than in the color belt group, as presented in Table 1, Table 2 and Table 3. The average overall duration of practice was not significantly different between males and females. There was no significant difference between men in SOS, BUA and SI. The size of the measured parameters was influenced by the level of advancement in martial arts (black/colored belt) (Table 2) and gender (Table 3). No significant differences in the order of weekly PA spent in dojos were found.

Table 4 compares the results of the current study with data available in the peer-reviewed literature.

## 4. Discussion

### 4.1. Basic Considerations

Osteoporosis, a common chronic disease, affects bone strength. The disease is more prevalent in the elderly [1,2,3,5,70]. Its significant decrease in bone strength is a significant risk factor for fractures. Osteoporosis is not always the result of bone loss, but it can also occur when an individual’s optimal peak bone mass is not reached by adulthood [6,8,27,71]. Such an effect can contribute to maintaining bone health during an individual’s life. During the developmental period, childhood, adolescence, and early adulthood, achieving the maximum peak value of bone strength is assumed to be crucial for preparing the bones for later years [72]. After the plateau of its density and strength, the mature bone resorbs [2,63,73], and finally, it may lead to osteoporosis [1,3,6,7]. This process may proceed slower or faster depending on several influential factors. PA significantly influences BMD and bone strength and slows resorption [15,17,20,42,72,74,75]. Childhood and adolescence are responsible for bone acquisition and achieving 90% of peak bone mass and strength [8]. Bones in this period of life respond positively to PA [76,77,78,79]. Once peak bone mass is reached, BMD stabilizes; after the age of 40, a gradual decline may begin and accelerate with age and the menopause in women [80]. Optimally reaching a high peak of bone strength (bone mass), with other parameters unchanged, is considered a protecting factor against osteoporosis later in life [6,8,27]. Although bone strength is currently included in the definition of osteoporosis, a direct measurement of bone strength in a non-invasive and non-destructive way is not yet available. Physics and biomechanical studies of bones have shown that ultrasonic parameters enable the determination of bone strength properties in the form of Young’s modulus. In addition, compression modulus, bone strength, and flexibility were significantly associated with SOS [62,63,81]. BUA and SOS measurements of intact calcaneus correlated significantly with human cadaveric calcaneal bone biomechanical parameters in vitro [82]. QUS has been recognized as an osteoporosis screening tool [9]. BUA and SOS are basic measurements for QUS. Additional QUS indicators are also used, which are derived from basic measurements. These include SI, quantitative ultrasonic index (QUI), and estimated BMD (eBMD). In upper limb QUS, amplitude-dependent SOS (AD-SOS) measurements are made across the proximal phalanges of the hand [83,84,85]. Additionally, measured parameters characterize the graphic waveform (rapid wave amplitude, signal dynamics, and bone transmission time (BTT)) [83,84]. Phalangeal AD-SOS of the hand can be helpful for good separation between the reference population’s age- and sex-matched control group and the clinically diagnosed osteoporosis cases with hip fractures [86].

Bone ultrasonometry is an appropriate quantitative method to distinguish between healthy individuals and patients with osteoporosis. Bone QUS can be performed quickly and assesses bone quantification, particularly emphasizing bone strength (lower limb bones). Normal values for a calcaneus ultrasound were studied by several authors [12,68,87,88,89,90,91]. Measurements revealed significantly higher results of QUS parameters than observed for the female and male normative populations [12]. Quantitative ultrasound bone parameters may represent the bone strength of the lower extremity. Higher QUS parameters mean higher bone strength remains. Lower values indicate higher proximal femur fracture risk [4,11,92,93,94]. Changes in lifestyle by introducing regular PA, quitting smoking, and using dietary supplementation products may prevent osteoporotic fractures [95,96,97]. Several papers documented the effects of general or specific physical training in males and females assessed by safe quantitative ultrasonometry measurements [12,13,15,16,98,99,100,101,102]. Calcaneal QUS measurements were performed by several researchers [49,103,104]. Many researchers have focused on preventing and treating osteoporosis in women due to its higher incidence. A previous study confirmed that QUS measurements at the calcaneus might provide information comparable to DXA examinations at the femoral neck and lumbar spine in post-menopausal women [105]. Several papers recommend PA as the preventive modality of lifestyle [30,43,106,107]. Current osteopenia treatment guidelines include optimal exercise regimens for attenuating BMD loss, and improving poor balance and muscle strength to address other fracture-related risk factors. Pluskiewicz et al. [68,91,108] measured upper extremity phalanges. They also performed a cross-sectional study to assess the ability of quantitative ultrasound at the calcaneus to discriminate between fractured and unfractured men, fracture probability, and the relationships of ultrasonic parameters with age and body size [69]. SOS = 1517.5 m/s and BUA = 114.0 dB/MHz were found in unfractured men, and SOS = 1492.6 m/s and BUA = 106.1 dB/MHz in fractured men. They confirmed the ability of quantitative ultrasound measurements at the calcaneus to discriminate between fractured and healthy males. A few studies were conducted to find the average quantitative ultrasound measurements of the calcaneus among both sexes’ normal subjects [12,68,69]. In terms of the QUS of the calcaneus, a cross-sectional study of 964 subjects [68] showed that SOS and BUA average values were significantly higher in healthy men (1517.5 +/− 35.3 m/s, 114.0 +/− 13.3 dB/MHz) than in healthy women (1511.1 +/− 25.6 m/s, 108.7 +/− 9.5 dB/MHz). The ultrasound variables had higher values in osteoporotic men (SOS = 1492.6 +/− 24.6 m/s, BUA = 106.1 +/− 11.6 dB/MHz) in comparison to osteoporotic women (SOS = 1490.4 +/− 19.5 m/s, BUA = 103.2 +/− 8.6 dB/MHz). Calcaneal QUS measurement similarly differentiates healthy male/female and osteoporotic populations.

### 4.2. Skeletal Status Assessed with QUS and PA and Non-Combat Sports

The effects of different sporting activities may be different. Some sports and activities are classified as weight-bearing and non-weight-bearing sporting activities of different intensities on bone mineral accrual in adolescence and early adulthood. The discussion about other sports activities or martial arts is primarily heterogeneous. Young males playing soccer had a better calcaneus than those who participated in cycling or swimming [20,109]. Female gymnasts gained significantly higher bone density than controls [42]. No significant differences were found in QUS parameters between swimmers and controls (both male and female). Male soccer players reported significantly higher QUS values than controls [110,111]. Weight-bearing PA and higher bone ultrasound parameters are described in the literature. Mixed weight-bearing programs that include jogging, walking, and stair climbing consistently improve hip BMD in older adults [112] and are considered the optimum type and level of PA. Gaudio et al. [111] found significantly higher QUS values in competitive footballers than in the controls. However, the stiffness index (%) differences were not spectacular (96.3 ± 4.1 vs. 93.0 ± 3.5 (*p* < 0.05)). It was hypothesized that competitive soccer players [110] have less contralateral variability in QUS measurements than normally active individuals due to football’s bilateral impact loading characteristics. Players had significantly higher BUA and SOS mean values (30.6% and 45.5% higher mean BUA and 3.38% and 3.87% higher mean SOS values for the right and the left heels, respectively) than those in non-athletic controls (*p* = 0.0001). Significant side differences were found in the mean values of BUA in the non-athletic subjects (*p* = 0.001) but not in the footballers (*p* = 0.538). The football players showed slightly higher BUA and SOS mean values on the left than those on the right. While the number of football players who showed higher BUA values on either side was similarly distributed, a significantly greater number of normally active males showed higher BUA values on the right side. The authors concluded that the high-impact loading of both calcanei in football is reflected by the acoustic parameters of bone, mainly BUA, in terms of overall enhanced bone properties and improved bone properties on the non-dominant side. Brahm et al. [113] compared endurance runners with BMD controls at various sites, peripheral QUS, and biochemical markers of bone metabolism. Thirty moderate endurance runners (32, from 19 to 54) were compared to an age- and sex-matched control population. In runners, calcaneus QUS showed higher BUA (9.2%; *p* = 0.002) and SOS (3.1%; *p* = 0.0001) than controls. In the control group, BMD was associated with follow-up ultrasound measurements at all sites, but this relationship was not observed in runners. In other sports, Yung et al. [114] found no significant differences in BUA and VOS (*p* > 0.05) between the dominant and non-dominant heel. They reported significantly higher BUA, VOS, and SI scores (*p* < 0.05) in soccer groups (137 +/− 4.3 dB/MHz; 1575 +/− 56 m/s; 544.1 +/− 48.4) and dancers (134.6 +/− 3.7 dB/MHz; 1538 +/− 46 m/s; 503.0 +/− 37.0). In the group of swimmers (124.1 +/- 5.1 dB/MHz; 1495 +/− 42 m/s; 423.3 +/− 46.9) and the control group in the sitting position (119.9 +/− 6, 1 dB/MHz; 1452 +/− 41 m/s; 369.9 +/− 46.4) measurements were lower. Studies have shown a significant linear increase in all QUS parameters with weight-bearing and high-impact exercise (*p* < 0.05). Female and male dancers (aged 19–36) and 100 non-athletic age- and gender-matched subjects were tested using calcaneus QUS [74]. Significant differences were found in all QUS parameters between the dancers and the control group. For each heel (right and left) of the dancers compared to the control group, the mean values were 1.1% QUI, 0.6% BUA, 2.5% SOS, and 1.1% eBMD. For female dancers, the mean values were higher than those of the female controls: QUI 0.3%, SOS 1.1%, eBMD 0.4%, and no difference in BUA. Henriques-Neto et al. [21] assessed the influence of vigorous physical activity (VPA) on QUS in 412 participants (221 girls) aged 10-18. They observed that in boys, handgrip mediated the associations of VPA with the speed of sound on the third distal radius (R-SoS). In contrast, the speed at 20 m and handgrip mediated the relationship of VPA with the speed of sound on the tibial midshaft (T-SoS). Handgrip, speed, and fat mass (%) in boys and cardiorespiratory fitness in girls mediate the relationships between VPA and bone health assessed by QUS. Promoting muscular and cardiorespiratory fitness and decreasing fat mass through VPA in adolescents may be essential strategies to improve bone health. Dib et al. [115] found no influence of PA on bone status. Hervas et al. [71] aimed the study at understanding the modifiable factors that improve and maximize peak bone mass at an early age. They identified moderate-to-vigorous PA and physical fitness as the strongest predictors of bone SI, measured by quantitative ultrasonometry in young university students (18–21 years). SI was closely related to vigorous PA in males and the steps per day number in females. The authors of this study identified predictors of bone status in each sex and indicated that muscle and bone interrelate with PA and fitness in young adults. PA (approximately 20 h per week) significantly influenced young females’ maximal peak bone mass values [116]. There were significantly higher values of SOS—25 m/s (*p* < 0.0005), BUA −5 dB/MHz (*p* < 0.018), and SI—11% (*p* < 0.0007) in the high activity group than in the sedentary group. The results strongly suggested that PA (approximately 20 h per week) significantly influences maximal peak bone mass values. PA in post-menopausal women, especially several times a week, significantly influences bone health [105]. Bolanowski et al. [117] assessed the ultrasound properties of the skeletal status (amplitude-dependent speed of sound—Ad-SoS) measured at hand phalanges in adolescent girls regarding the influence of pubertal status and the level of physical fitness. The statistically significantly higher results of the Ad-SoS were found in girls presenting the highest level of physical fitness than in girls with a minimal level of physical fitness. Chwałczyńska et al. [85] assessed hand phalanges QUS. Their study showed that Ad-SoS and shoulder muscles strength in boys were positively correlated. No significant differences in any calcaneal QUS parameters were found when measuring the non-dominant calcaneus between any swimmers groups versus controls [76,77,118]. Mentzel et al. [119] reported higher SOS values in athletes than wrestlers, basketball players than fencers, basketball players than wrestlers, and gymnasts compared with judokas sports players. Organized recreational activity effectively maintained bone status among perimenopausal and post-menopausal women [120,121,122]. Karlsson et al. [123], in soccer players, found significantly better BMD measurements performed at the spine, total body, and hip and QUS calcaneal parameters than in controls, irrespective of training frequency. Cheng et al. [124] presented a BMD of the calcaneus evaluated by single photon absorptiometry (SPA) and a QUS that were better in jumpers than in other athletes and controls, while QUS parameters did not show any differences. Calcaneal QUS measurement variables were significantly better in those who regularly participated in sports, either currently or in the past. Six months of exercising were the most significant determinant influencing QUS variables in Finnish men aged 18–20 [103]. Bone quality is rarely tested among martial artists [18,52,53,55]. Studies on Okinawa Kobudo/Karate Shorin-Ryu practitioners have not been published so far. The combined Kobudo/Karate practice was assumed to lead to a high musculoskeletal system activation, which explains the study results. All Kobudo practitioners also practiced karate, considered a sport with a positive influence on skeletal status, with the most significant benefits occurring in adults [50,52,53]. The current study is the first in the literature showing skeletal changes in this group’s lower extremities. Adolescents and elderly groups were studied more frequently to find bone density changes in martial arts practitioners. Papers have rarely focused on middle-aged groups. Several reviews have shown that various martial arts exercises can improve bone quality, balance, and falls risk in adults and older adults [43,44,48,125,126]. Gravitational exercises play a crucial role in martial arts performance. Studies were more frequently conducted in older groups who practiced tai chi, but younger groups were studied who practiced other martial art activities [22,43,45,46,47,48,73,106,127,128,129,130,131,132,133,134]. The beneficial role of exercise in improving BMD, muscle strength, and balance has been documented mainly in younger populations. These results may not apply equally to older populations with limited ability to perform high-intensity exercise. Studies that investigated the effectiveness of the increasingly popular mind–body tai chi practice on BMD, postural balance, and fracture risk suggested beneficial effects in women with low bone density. Short-term training was described as beneficial for bone density. How bone density changes in a group of people who have to discontinue their regular or almost daily exercises remains unanswered. Groups with a higher osteoporotic fracture risk may gain benefits of tai chi practice that can see an improvement in balance but rarely a rise in BMD. Direct fall risk reduction and improved postural control, reduced fear of falling, static and dynamic factors, skeletal muscle strength, and flexibility in daily activities are among the tai chi benefits reported in systematic reviews of randomized trials [43,106,125,129,131,132,135,136,137]. It was found that tai chi can significantly reduce falls or hip fracture costs. Most studies have focused on the elderly population. It is stressed that these results are important for older women, and tai chi can be practiced safely in the later stages of life. Research on the direct impact of tai chi on BMD is more limited and less conclusive. The studies available in the literature indicate that long-term tai chi practice allows for a higher BMD (compared to controls) and a slower decline in BMD in post-menopausal women. The beneficial effects of tai chi exercise on the BMD of the proximal femur were equivalent to 12 months of resistance training. Tai chi is considered a safe PA for women after menopause. The problem of falls affects approximately 30% of people over 65 living in the community yearly [44,125,137,138,139]. Gillespie et al. [125] assessed and validated the effects of tai chi interventions to reduce falls among elderly people living in the community. However, the reduction in the rate of decline only bordered on statistical significance. Wayne et al. [48] affirmed the value of longer-term regular tai chi practice to achieve clinically detectable relevant effects on BMD loss and reduce the fall risk in the study population. Tai chi may benefit balance, falls, and non-vertebral fractures. However, only Lee et al. [106] concluded in their systematic review that the evidence for tai chi in the prevention or treatment of osteoporosis is not convincing. In 2007 Wayne et al. [129] evaluated the evidence for tai chi as an intervention to reduce the rate of bone loss in post-menopausal women. They found that cross-sectional studies suggested that long-term tai chi practitioners had higher BMD than age-matched sedentary controls and had slower rates of post-menopausal BMD decline. No adverse effects related to tai chi were reported in any trial. They concluded that there was limited evidence that tai chi might be an effective, safe, and practical intervention for maintaining BMD in post-menopausal women. In a randomized controlled trial by Woo et al. [130], there was less BMD loss at the total hip in tai chi women than in the control group and no effect in men. The modest beneficial effect of tai chi on musculoskeletal health did not translate into better clinical outcomes. Qin et al. [128] examined 99 healthy post-menopausal women to evaluate the potential benefits of regular tai chi chuan exercise on BMD and neuromuscular function. The tai chi group showed, overall, an approximately 7% higher BMD at all measurement sites, with a significant difference found at the spine, greater trochanter, and Ward’s triangle of the proximal femur, and significantly greater quadriceps strength. They concluded that regular exercise might be associated with higher BMD and better neuromuscular function in early post-menopausal women. Chan et al. [127] conducted age-matched and randomized prospective intervention for early post-menopausal women. After the 12-month follow-up study, BMD measurements revealed a general bone loss in all study participants at all measured skeletal sites, but with a reportedly slower rate in the tai chi group. The case–control study by Qin et al. [73] evaluated the potential benefits of regular tai chi chuan exercise on the weight-bearing bones of post-menopausal women using DXA BMD in the lumbar spine and proximal femur and in the distal tibia using multislice peripheral quantitative computed tomography (pQCT) at baseline and follow-up 12 months later. They concluded that regular tai chi chuan exercise might help retard bone loss in the weight-bearing bones of post-menopausal women. Combat sports such as judo, kung fu, karate Kyokushinkai, and traditional boxing may also influence skeletal status. Tsang et al. [140] mentioned a possible link between the practice of Chinese martial arts (kung fu) and beneficial physiological benefits (e.g., aerobic capacity and bone density).

The results of some studies suggested that the type of sports activity may be an essential factor in achieving a high peak bone mass and reducing osteoporosis risk. Andreoli et al. [55] investigated the effects of different high-intensity activities on BMD and muscle mass in highly trained judo, karate, and water polo athletes, who all competed at national and international levels. Age-matched non-athletic individuals served as the control group. They found a significantly higher total BMD in either judo or karate athletes. Tenforde and Fredericson [18] reviewed the literature evaluating sports participation in young athletes aged 10–30. High-impact sports (e.g., gymnastics, hurdling, judo, karate, volleyball, and other jumping sports) or odd-impact sports (e.g., soccer, basketball, racquet games, step aerobics, and speed skating) are associated with higher bone mineral composition and BMD, and enhanced bone geometry in anatomic regions specific to the loading patterns of each sport. It was postulated that judo athletes gain bone density in the same way as other martial artists. Nasri et al. [53] conducted a study to investigate the correlation between bone parameters and grip strength in hands, explosive legs power, and hormonal parameters; second, they identified the most determinant variables of BMD among adolescent combat sports athletes. They compared the results of 50 combat sports athletes aged 17.1 ± 0.2 years with 30 sedentary subjects matched for age, height, and pubertal stage. All BMD measurements were more remarkable in athletes than in the sedentary group (*p* < 0.01). The GS and ELP showed higher values in athletes than in the sedentary group (*p* < 0.01). The grip strength of the non-dominant arm was the best predictor of BMD measurements. The osteogenic effect of combat sports practice, especially judo and karate Kyokushinkai, was also confirmed among judoists. Prouteau et al. [56] found an increased bone formation rate that may protect judo athletes from alterations in a bone metabolic balance associated with weight cycling. Unique papers presented QUS studies of phalanges among karate exercisers. Drozdzowska et al. [50] examined 226 males at an average age of 25.6 regularly exercising karate for an average of 5.2 years approximately three times a week, using DBM Sonic 1200 (IGEA, Italy) for the bone status of the hand’s phalanges assessment. In the current study, long-term black belts had a training period of 22.8 years; color belts were only eight years old. Observations confirmed that in the group of black belts, the QUS scores were also significantly higher than in the color belts. Even though significantly higher values of QUS parameters were found in the group of black belts (instructors) active daily, the colored belts also presented QUS results higher than the population average. The results suggest that the differences increased and stabilized after age 35, which shows that continuous training leads to a further increase in all parameters of the QUS scores. Long-term exercise with an average weekly activity of 10 h per week spent in the dojo leads to high quantitative bone QUS measurements. The positive effects of various martial arts exercises have been described independently of martial arts schools and styles. The critical issues for gaining bone mass relate to exercise performed frequently during the week, almost all year round. Thus, a constant difference exists between karatekas and the control group after age 35. This observation can be considered particularly favorable since fractures mainly occur in the elderly. In some studies, male bone measurements were only performed with a device suitable for phalanges [50,141]. A case–control study by Drozdzowska and Pluskiewicz suggested that the higher increase was connected with a high impact of PA, exceeding the standard level observed in karate. Drozdzowska et al. [50] assessed the influence of regularly exercised karate on the skeletal status measured (Ad-SoS (m/s)_ as quantitative ultrasound parameters at hand phalanges. Ad-SoS, T-score, and Z-score were significantly higher in the examined karate practitioners than in the control ones. The longer duration, higher frequency, and earlier exercise began positively affected skeletal health, with the most significant benefit being seen in adults. Regular karate training [142] was positively associated with the results of the QUS measurements at hand phalanges in exercising females. Its impact is most strongly pronounced in prepuberty and adulthood. The results of Ad-SoS obtained in karatekas were generally higher than in controls, with significant differences for prepubertal girls (1966.2 (SD 46.2) vs. 1942.7 (SD 38.4); *p* < 0.05) and for adult women (2124.4 (SD 48.0) vs. 2105.3 (SD 54.0); *p* < 0.05). A similar study in Brazil [143] compared phalangeal QUS bone mass (AD-SoS and BTT) in young karate practitioners with a control group. Women in the control group had higher AD-SoS values (m/s and Z-score) than karatekas. For the BTT Z-score in both groups, it was shown that the bone mass of men practicing karate was more often appropriate. In practicing women, age and weight were independent predictors of AD-SoS (R2 = 0.42) and BTT (R2 = 0.45), respectively. Among male karate practitioners, age was 26% of AD-SoS variance, and height was 36% of BTT variance. It turned out that BTT was the most adequate and appropriate parameter for the assessment of bone tissue in karatekas due to its greater sensitivity. Irrespective of gender, children, and adolescents, it was shown that karate athletes had a higher bone mass index than the control group [22]. Significant changes in bone health come from high-impact sports (football, gymnastics and dance, and martial arts). However, it has not been shown that there are such positive effects on the bone tissue in swimmers and cyclists compared to the control group. Higher QUS parameters were associated with weight-bearing and high-impact exercise. Soccer players and dancers had significantly greater BUA, VOS, and SI than swimmers and the sedentary control group [114]. Soccer leads to the statistically more significant QUS ultrasound parameter SI than swimmers, cyclists, and controls at baseline [109]. Ving-Tsun (VT) training has also been shown in older adults to lead to significantly higher bone strength than age-, sex-, and height-matched senior controls [133,144,145]. Studies of the effect of tai chi [22] on markers of bone turnover and calcaneal QUS did not provide sufficient evidence to support an effect. The current study focused on skeletal changes in the lower extremities in martial artists practicing Kobudo and karate, which may also mean longer or more frequent barefoot activity. Quantitative ultrasonometry was used to evaluate differences between subjects who extensively practiced and more or less new beings who practiced for short periods—significant differences were observed between the studied long-term, extensive practitioners versus the short-term practitioners. The duration of practice expressed by belt color was easy to apply. It proved the positive influence of physical training on skeletal status. The number of training hours per week could play a similar role. Exercising duration in various papers is measured in months. Long-term effects are rarely investigated. The phenomenon of the study group consists of voluntary physical training participation for many years. Unlike many competitive and Olympic sports, the described martial arts style is often practiced for a long time. The age range of “Black” belts was significantly higher within the studied group than in the less advanced group. Among the holders of master’s degrees (black belts) with a much longer training experience, significantly higher bone quality was observed. The achieved results in this study group are far beyond the norm range for age-matched sedentary subjects. Non-smoking behavior observed in the study group may play an essential role in achieving significantly high SOS, BUA, and stiffness values in addition to long-term exercise. Judokas and wrestlers showed a significant positive correlation between heel BUA and activity level. There were significant differences between badminton players and gymnasts, basketball players and fencers, and judokas and gymnasts. Numri-Lawton et al. [42] studied gymnasts and revealed that they were smaller and lighter than controls but still had significantly higher QUS. The QUS devices described in the articles used to test bone ultrasound parameters do not examine bone tissue in the same places. Examination of the peripheral skeleton of the upper limb (phalanges of the hand) may be controversial because it does not belong to the supporting skeleton of the entire body weight. In addition, different devices give measurements in different parameters. For this reason, the output values measured with them cannot be directly compared. The calcaneus examinations performed with various devices can be carried out in a water bath through water-filled membranes or connectors, ensuring direct contact with the examined tissues. Another limitation of this review is the relatively broad age of the study participants. Details of sports measurements for sports history, duration, and other included physical activities were heterogeneous, and sometimes the methods of recording and confirming details could have been more apparent. For example, the duration of regular sporting activity before inclusion in each study should have been reported. Average weekly sports training regimes ranged from a minimum of a few hours to a maximum of 27 h, and levels of participation in non-elite sporting activities between studies were not directly comparable. Interstudy comparisons cannot be made because, in the selected studies, the criteria for selecting participants and controls needed to be more consistent between studies. Different ultrasound methods and devices obtained the measurement results.

Than et al. [146] found motoric feature differences in lower limbs after barefoot exercises. Significant differences in the QUS parameters were observed only between the group of men and the group of women. It was assumed that the critical factor influencing the quantitative changes in the bone tissue of the calcaneus is long-term PA with barefoot loading and not in sports shoes with shock absorbers.

This study has some limitations. First, the sample size was relatively small, especially in the colored band group. Although the test’s statistical power was not initially high, significant differences in the results were demonstrated.

Second, study participants were not randomized, and the evaluator was not blinded, which may have introduced bias and biased results. In addition, due to the relatively distant time of the study and the difficult period of the pandemic for public health, a repeat study could indicate further observations, especially whether the results show a consistent trend.

The QUS method seems appropriate to observe changes in the quantitative measure of bone tissue. Systematic reviews of studies usually address research on sporting activities with bone health using the QUS. Many studies have shown that calcaneal ultrasonometry is used worldwide to assess bone structure and strength in assessing the risk of osteoporotic fractures when DXA, the gold standard diagnostic tool, is unavailable. However, it should not be a diagnostic tool [48]. The positive features of calcaneal ultrasound measurements are that they involve no risks or harm and are cost-effective, convenient, painless, and radiation-free examinations that are easy to use and take only a few minutes. Studies have shown that quantitative ultrasonometry helps assess changes in skeletal health due to exercise in all age groups and as a research tool. The consistency of the results suggests that the ultrasound examination can be successfully used in studying bone tissue in sports.

## 5. Conclusions

Based on these studies, the osteoporosis prevention potential of Okinawa Kobudo/Shorin-Ryu Karate seems clear as long-term martial arts training significantly affects QUS parameters of the calcaneus. Long-term martial arts practitioners achieved results significantly exceeding the norms for age groups. The significantly higher bone density among black belt holders may be cumulative due to the usually longer daily dojo and barefoot activity. The good general health and no-smoking behavior declaration also look sound in this study.

More cohort studies are needed to predict when martial arts exercise becomes osteoprotective. It remains to be determined whether the sport that leads to the most noticeable improvement in ultrasound bone has been identified. Further research should include a larger randomized trial, blinded evaluations, and documentation to confirm beneficial effects. An additional general physical fitness evaluation method would probably be advisable to draw more general results.

Further studies using non-invasive methods of bone quantification are needed to determine the specific determinants of osteoporosis prevention through physical activity and sport, particularly the duration of the activity, the magnitude of loads, and other factors related to practicing sports.

## Figures and Tables

**Table 1 ijerph-20-02708-t001:** Descriptive statistics show the mean values of variables characterizing the participants.

Variable	Mean	Minimum	Maximum	Std. Dev.
Age	36.4	20	68	10.72
Years of practice	15.9	1.0	53.0	11.65
SI	115.4	87.3	154.2	17.53
SOS	1608.4	1540.5	1725.8	44.76
BUA	127.9	98.2	154.9	11.44
YoungAdult%	115.4	87.3	154.2	17.5
T-Score	0.9	−0.79	3.4	1.09
Z-Score	1.1	−0.75	3.7	1.15
Age-Matched%	119.6	87.8	173.2	20.14

**Table 2 ijerph-20-02708-t002:** The breakdown table of descriptive statistics of mean values and standard deviations of variables characterizing the black and color belts in the study group.

Variable\Belt	Black (Mean)	Black (STD)	Color (Mean)	Color (STD)	All (Mean)	All (STD)
Years of practice (years)	22.8	10.9	7.8	5.8	16.0	11.7
SI	121.7	18.8	107.9	12.7	115.4	17.5
SOS (m/s)	1620.6	49.2	1593.8	34.5	1608.4	44.8
BUA (dB/MHz)	132.3	9.8	122.8	11.3	128.0	11.4
YoungAdult%	121.7	18.8	107.9	12.7	115.4	17.5
T-Score (SD)	1.4	1.2	0.5	0.8	1.0	1.1
Z-Score (SD)	1.6	1.2	0.6	0.8	1.2	1.2
Age-Matched%	127.9	21.5	109.8	13.1	119.6	20.1

**Table 3 ijerph-20-02708-t003:** The breakdown table of descriptive statistics of mean values and standard deviations of variables characterizing the females and males in the study group.

Variable/Sex (M/F)	M (Mean)	M (STD)	F (Mean)	F (STD)
Age	37.72973	10.79261	29.28571	7.47695
Years of practice	16.62162	12.32962	12.57143	6.67975
SI	116.2836	18.67943	110.8992	9.00148
SOS	1610.009	47.78311	1600.017	23.75996
BUA	128.5883	11.6732	124.6752	10.30149
YoungAdult%	116.2836	18.67943	110.8992	9.00148
T-Score	1.017726	1.167464	0.681201	0.562592
Z-Score	1.239812	1.226614	0.723421	0.556512
Age-Matched%	121.1548	21.36752	111.647	8.94256

**Table 4 ijerph-20-02708-t004:** Average values and standard deviations of SOS and BUA measured in the study group and values presented in the literature as characteristic for healthy, fractured, and osteoporotic subjects. * Comparison was based on measurement values presented in the literature [68,69].

Group/Variable	SOS [m/s]	SD [m/s]	BUA [dB/MHz]	SD [dB/MHz]
Kobudo practitioners	1608.4	+/−49.2	127.9	+/−9.8
Females	1610.0	+/−44.76	124.6	+/−10.3
Males	1608.4	+/−25.7	127.9	+/−11.4
Black belts	1620.6	+/−34.5	132.2	+/−11.3
Color belts	1593.7	+/−44.7	122.8	+/−11.4
unfractured men *	1517.5	+/−35.3	114.0	+/−13.3
fractured men *	1492.6	+/−24.6	106.1	+/−11.6
healthy men *	1517.5	+/−35.3	114.0	+/−13.3
healthy women *	1511.1	+/−25.6	108.7	+/−9.5
osteoporotic men *	1492.6	+/−24.6	106.1	+/−11.6
osteoporotic women *	1490.4	+/−19.5	103.2	+/−8.6

## Data Availability

Data supporting reported results can be available on-demand for the readers.

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
