# Peer review of "Quantitative Ultrasound Examination (QUS) of the Calcaneus in Long-Term Martial Arts Training on the Example of Long-Time Practitioners of Okinawa Kobudo/Karate Shorin-Ryu"

_ijerph, 2023, doi:10.3390/ijerph20032708_

Round 1

Reviewer 1 Report

 Review

The article addresses an interesting and pertinent topic. In fact, martial arts and combat sports in general involve high-impact physical activity. The impact (weight-bearing exercise) stimulates the activation of osteoblasts (cells involved in bone growth) promoting an overall improvement in bone mineral density and thus preventing osteoporosis. The present study confirms its effect through the constant practice of Kobudo/Karate, with the highest effect being evidenced by black belt practitioners, that is, the improvement in bone health is related to longer practice time.

I suggest a little revision and corrections according to specific comments made in the manuscript.

Author Response

Respectful Reviewer I:

The authors are thankful for carefully reading and pointing out every issue requiring corrections.

The erroneous words and sentences were corrected according to the reviewer's comments.

Suggested citations were added.

Reviewer 2 Report

I greatly appreciate the topic and research done by the authors.
The most important for me are the methodological improvements:
Justify the sample size and selection.
It is worth providing specific inclusion and exclusion criteria for the study.
In what period were the studies carried out? I appreciate the reference to ethical standards and the positive assessment of the committee, but its date suggests that the research is from 11 years ago! This does not disqualify this text, but it raises the question whether, due to the lifestyle and changes in public health in recent years, are they up to date? If so, please justify it thoroughly!
I appreciate the extensiveness and multithreading of the discussion, but the conclusions are far too short, moreover, few limitations are given, and these are certainly the size of the sample and the unrepresentativeness of the study, as well as (probably) the date of the study.

Author Response

Respectful Reviewer II:

The authors are thankful for carefully reading and pointing out every issue requiring corrections.

It is our pleasure to read the reviewer's appreciation of the study.

The authors are thankful for the methodological comments concerning the sample size and selection. A broader explanation is added in the material and method chapter.

The authors are thankful for the specific inclusion and exclusion criteria comment—the explanation of the enrollment of the participants for the study is added.

The authors are thankful for pointing out the study period. The study was performed ad hoc just after obtaining the Bioethical Committee's approval and the confirmation of the advertisement of the International Stage in Warsaw. Indeed the study is old, and the final manuscript took a long time.

The authors are thankful for pointing out the size of the conclusion, and a few remarks were added to the Conclusions chapter.

The descriptions were extended.

Round 2

Reviewer 2 Report

The authors took into account my comments.
I still miss a specific time period for the study: from X to Y

Author Response

The authors are thankful to Reviewer 2 for accepting most of the corrections.

The exact dates are provided in the revised manuscript.

Kind regards

Sincerely

Wojciech Glinkowski

on behalf of the co-authors
